Vegetation responses to season of fire in an aseasonal, fire-prone fynbos shrubland

Kraaij Tineke tineke.kraaij@mandela.ac.za 1
Cowling Richard M. 2
van Wilgen Brian W. 3
Rikhotso Diba R. 4
Difford Mark 2
1 School of Natural Resource Management, Nelson Mandela University , George , South Africa
2 Botany Department, Nelson Mandela University , Port Elizabeth , South Africa
3 Centre for Invasion Biology, Department of Botany and Zoology, University of Stellenbosch , Stellenbosch , South Africa
4 Garden Route Scientific Services, South African National Parks , Knysna , South Africa
Daehler Curtis
Electronic publication date: 2017 Aug 10
Publication date: 2017
Volume: 5
Electronic Location ID: e3591
Received 2017 Feb 13; Accepted 2017 Jun 28
Copyright: ©2017 Kraaij et al.
Copyright year: 2017
Copyright holder: Kraaij et al.
License: This is an open access article distributed under the terms of the Creative Commons Attribution License, which permits unrestricted use, distribution, reproduction and adaptation in any medium and for any purpose provided that it is properly attributed. For attribution, the original author(s), title, publication source (PeerJ) and either DOI or URL of the article must be cited.
License URL: https://creativecommons.org/licenses/by/4.0/

Keywords: Fire season, Post-fire recruitment, Protea, Germination, Prescribed burning, Cape Floral Kingdom, South Africa, Leucadendron, Seed planting experiment

Funding: South African National Parks Nelson Mandela University Council for Scientific and Industrial Research (South Africa) South African Environmental Observation Network This work was supported by South African National Parks, the Nelson Mandela University, the Council for Scientific and Industrial Research (South Africa), and the South African Environmental Observation Network. The funders had no role in study design, data collection and analysis, decision to publish, or preparation of the manuscript.

==============================
Season of fire has marked effects on floristic composition in fire-prone Mediterranean-climate shrublands. In these winter-rainfall systems, summer-autumn fires lead to optimal recruitment of overstorey proteoid shrubs (non-sprouting, slow-maturing, serotinous Proteaceae) which are important to the conservation of floral diversity. We explored whether fire season has similar effects on early establishment of five proteoid species in the eastern coastal part of the Cape Floral Kingdom (South Africa) where rainfall occurs year-round and where weather conducive to fire and the actual incidence of fire are largely aseasonal. We surveyed recruitment success (ratio of post-fire recruits to pre-fire parents) of proteoids after fires in different seasons. We also planted proteoid seeds into exclosures, designed to prevent predation by small mammals and birds, in cleared (intended to simulate fire) fynbos shrublands at different sites in each of four seasons and monitored their germination and survival to one year post-planting (hereafter termed ‘recruitment’). Factors (in decreasing order of importance) affecting recruitment success in the post-fire surveys were species, pre-fire parent density, post-fire age of the vegetation at the time of assessment, and fire season, whereas rainfall (for six months post-fire) and fire return interval (>7 years) had little effect. In the seed-planting experiment, germination occurred during the cooler months and mostly within two months of planting, except for summer-plantings, which took 2–3 months longer to germinate. Although recruitment success differed significantly among planting seasons, sites and species, significant interactions occurred among the experimental factors. In both the post-fire surveys and seed planting experiment, recruitment success in relation to fire- or planting season varied greatly within and among species and sites. Results of these two datasets were furthermore inconsistent, suggesting that proteoid recruitment responses are not related to the season of fire. Germination appeared less rainfall-dependent than in winter-rainfall shrublands, suggesting that summer drought-avoiding dormancy is limited and has less influence on variation in recruitment success among fire seasons. The varied response of proteoid recruitment to fire season (or its simulation) implies that burning does not have to be restricted to particular seasons in eastern coastal fynbos, affording more flexibility for fire management than in shrublands associated with winter rainfall.

Introduction

Fires ignited by lightning are the dominant natural disturbance in the species- and endemic-rich fynbos shrublands (Myers et al., 2000) of the Cape Floral Kingdom (CFK), South Africa (Kruger & Bigalke, 1984; Seydack, Bekker & Marshall, 2007). Empirical evidence indicates that season of fire can affect species abundance and floristic composition in fire-prone Mediterranean-climate shrublands of the world (Bond, 1984; Enright & Lamont, 1989; Midgley, 1989; Domínguez, Calvo & Luis, 2002; Keeley, 2006; Moreno et al., 2011; Céspedes et al., 2012). Knowing how species respond to fire regimes (including fire season) is essential for ecologically sustainable management (Driscoll et al., 2010).

In the CFK, existing fynbos fire management protocols restrict burning to the late summer-early autumn period (Van Wilgen, 2009). These protocols are largely based on knowledge of the summer-autumn fire regimes in the west (Kruger & Bigalke, 1984; Van Wilgen, Richardson & Seydack, 1994) where the climate is Mediterranean, with cool, wet winters and warm, dry summers (Van Wilgen, 1984), and where post-fire plant recruitment is accordingly seasonally constrained (Van Wilgen & Viviers, 1985; Midgley, 1989). In contrast, rainfall in the eastern coastal part of the CFK occurs year-round (Schulze, 1965), weather conditions conducive to fires (Kraaij, Cowling & Van Wilgen, 2013a) and fire occurrence (Kraaij et al., 2013b) are largely aseasonal, and comparatively little is known about the effects of fire season on post-fire plant recovery (Van Wilgen, 2009).

Obligate reseeding shrubs are often particularly susceptible to population declines under fire regimes that do not provide favourable post-fire recruitment conditions. This is evident in shrubs regenerating from soil-stored seed banks in Californian chaparral (Keeley, 2006) and Mediterranean Basin maquis (Moreno et al., 2011; Céspedes et al., 2012), and from canopy-stored (serotinous) seed banks (almost exclusively members of the Proteaceae; hereafter ‘proteoids’) in Australian kwongan (Bell, 2001; Lamont, Connell & Bergl, 1991a; Enright et al., 1998) and South African fynbos (Jordaan, 1949; Bond, Vlok & Viviers, 1984). Proteoids release their whole seed bank post-fire (Lamont et al., 1991b) and the seeds are short-lived after release, mostly germinating during the first favourable period (Cowling & Lamont, 1987). Germination and early survival are critical phases in post-fire proteoid establishment (Bond, 1984; Midgley, 1988; Midgley, Hoekstra & Bartholomew, 1989; Mustart et al., 2012) while subsequent mortality of saplings is low (2–7% over a period of 4–6 years, Kraaij et al., 2013c). Inter-fire recruitment is rare and seedlings often die (Gill, 1975; Keeley et al., 2011). Extreme variation in post-fire recruitment is characteristic of proteoids (Van Wilgen & Viviers, 1985; Cowling & Gxaba, 1990), but in the Mediterranean-climate shrublands of the CFK and Australia, proteoid recruitment is generally higher after fires in summer–autumn than in winter-spring (Van Wilgen & Viviers, 1985; Cowling & Lamont, 1987; Bradstock & O’Connell, 1988; Enright & Lamont, 1989; Midgley, 1989).

Variation in recruitment success has been explained on the basis of: the size of pre-fire seed banks which vary with plant age (Le Maitre, 1990; Lamont, Connell & Bergl, 1991a; Kraaij et al., 2013c; Treurnicht et al., 2016), plant phenology (Jordaan, 1949), pre-fire plant density (Bond, Vlok & Viviers, 1984; Le Maitre, 1988; Cowling & Gxaba, 1990), pre-dispersal seed predation (Esler & Cowling, 1990), post-dispersal seed predation and decay (and thus duration of seed exposure between release and germination; Bond, 1984), the role of fire in breaking seed dormancy (Bond, Le Roux & Erntzen, 1990; Bond, Honig & Maze, 1999; Brown & Botha, 2004), climatic conditions favourable to germination, and the extent of post-germination mortality due to fungal pathogens, vertebrate and invertebrate herbivory, drought-stress and density-dependent thinning (Cowling & Lamont, 1987; Enright & Lamont, 1989; Midgley, 1989; Botha & Le Maitre, 1992; Mustart et al., 2012). Mechanisms accounting for effects of fire season, in particular, on recruitment success of obligate reseeding shrubs in Mediterranean-climate shrublands include (i) plant phenology (timing of seed production/dormancy/dispersal/predation in relation to timing of fire), (ii) interactions between fire season and fire intensity affecting the provisioning of germination cues, and (iii) the timing of climatic conditions favourable to germination (temperature stratification) and seedling survival (post-fire rainfall/desiccation) (Bond, Le Roux & Erntzen, 1990; Enright & Lamont, 1989; Moreno et al., 2011; Céspedes et al., 2012; Mustart et al., 2012).

Few studies have examined the effects of fire season on plant populations in fire-prone shrublands with a non-seasonal rainfall regime, where both fire season and intensity are variable (Bradstock, Williams & Gill, 2002; Van Wilgen, 2009). In eastern inland fynbos, Heelemann et al. (2008) established that plant phenology and thus, seed availability, do not impose seasonal constraints on proteoid recruitment (cf. Le Maitre, 1988), but that recruitment peaked after fires in autumn and spring, presumably coinciding with the bimodal peaks in rainfall. We aimed to determine whether post-fire recruitment seasonality in the eastern coastal (climatically more equable) part of the CFK differs from that in other parts of the CFK or whether recruitment success is unrelated to fire season. In addition to field surveys of recruitment success after fires in different seasons, our approach entailed a seed planting experiment aimed at identifying the mechanisms that determine post-fire recruitment seasonality. Finally, we explored the management implications of our findings to inform ecological fire management protocols in fynbos shrublands associated with a non-seasonal rainfall regime.

Materials & Methods

Study area

We focussed on the eastern coastal CFK, and in particular, the coastal (south-facing and thus cooler and wetter) slopes of the Outeniqua Mountains (east of the Touw River) and Tsitsikamma Mountains (33.80°S, 22.59°E–34.01°S, 24.26°E; Fig. 1). A large portion of the area occurs within the Garden Route National Park (Kraaij, Cowling & Van Wilgen, 2011).

Figure 1 Map of the study area.

Locations of the sites where post-fire recruitment surveys (multiple sites in close proximity may not be discernible at this scale; see also Table S1) and a seed planting experiment were conducted. The study area is divided into the Outeniqua and Tsitsikamma regions (a and b in insert, respectively), the Keurbooms River being the divide between these mountain ranges. The insert shows the location of the study area in relation to the Cape Floral Kingdom (CFK, grey-shaded) and South Africa.

Owing to maritime influence, the climate of the area is relatively equable (Schulze, 1965). Mean minimum and maximum temperatures range from 7°C and 19°C in June to 15°C and 26°C in January (Bond, 1981; Southwood, 1984). Rainfall occurs throughout the year, with 19% of annual rain falling during summer, 23% during autumn, 28% during winter, and 30% during spring (over the period 1993–2013 at the town of Plettenberg Bay, centrally positioned within the study area; Fig. 1). Mean annual rainfall increases eastwards, from 820 to 1,078 mm in the Outeniqua and Tsitsikamma Mountains, respectively (Bond, 1981; Southwood, 1984). The proportion falling as summer rain also increases eastwards (Schulze, 1965). While in the western-CFK, weather conditions suitable for fires dominate in the dry summer months (Van Wilgen, 1984), they are less seasonal in the eastern coastal CFK (Kraaij, Cowling & Van Wilgen, 2013a) with fires occurring any time of year (Kraaij et al., 2013b). Hot and dry katabatic berg winds in autumn and winter are associated with increased incidence, size and severity of fires (Kraaij, Cowling & Van Wilgen, 2013a) and their spread from the northern to the southern slopes of the coastal mountains.

The fire-prone and fire-dependent vegetation of the study area largely comprises montane sandstone fynbos (Rebelo et al., 2006). These are tall, medium-dense proteoid shrublands, with an ericoid-leaved shrub understorey (dominated by Ericaceae) and a prominent restioid (Restionaceae) component. Common overstorey proteoids are Leucadendron eucalyptifolium (Le), L. uliginosum (Lu), Protea eximia (Pe), P. mundii (Pm) and P. neriifolia (Pn) (nomenclature follows Rebelo, 2001). Flowering times are: Le, July–October; Lu, November–December; Pe, July–December; Pm, January–September; and Pn, February–November (Rebelo, 2001). However, there is a shift from winter-spring flowering to summer-autumn flowering both across and within lineages in the eastern-CFK (Cowling, 1987), where Pn flowers in summer and Le in spring (Heelemann et al., 2008).

Post-fire recruitment surveys

In natural vegetation with known recent histories of fire occurrence, we undertook one-off surveys (during 2007–2012) of recruitment success of overstorey proteoids (Le, Lu, Pe, Pm, Pn) within four years post-fire (1.9 ± 0.7 years, mean ± standard deviation). We counted the number of proteoid seedlings (post-fire recruits) in relation to the number of burnt parents (pre-fire population) within belt transects (2 m × 30 m). Proteoid seedling-parent counts are an established method for studying aspects of fynbos post-fire recruitment success (e.g., Bond, 1980; Bond, Vlok & Viviers, 1984; Van Wilgen & Viviers, 1985; Midgley, 1989; Cowling & Gxaba, 1990; Heelemann et al., 2008). Previous studies largely surveyed proteoid densities in 1 m2 quadrats spaced 5 m apart along lines, but belt transects (2 m in width; Treurnicht et al., 2016) provide for more time-efficient data collection, particularly given low proteoid densities. We surveyed 26 sites throughout the study area (Table S1; Fig. 1), each of which represented a particular fire or unique habitat (in terms of slope and/or aspect) within a fire. One or more proteoid species occurred at each site (Table S1) with site-species combinations providing 46 replicates for analysis. We surveyed 2–14 transects (recording all proteoid species present; Table S1) per survey site, aiming to record at least 30 parents and 30 seedlings per species at each site. However, at sites with low proteoid densities these numbers could not always be attained despite surveying more transects. Of the 46 site-species replicates sampled, six were winter burns, 20 spring, 12 summer and eight autumn burns. Fire return interval (∼pre-fire vegetation age) ranged from seven to 38 years, i.e., intervals where recruitment is unlikely to be constrained by seed shortages associated with juvenile proteoids (Kraaij et al., 2013c). We obtained rainfall figures (measured at the town of Plettenberg Bay) for a period of six months after each fire surveyed.

We combined the data of all transects within each site-species replicate and calculated for the latter the seedling-parent ratio as a measure of recruitment success. We were primarily interested in the effects of fire season (n = 46) on recruitment success, but considered other variables known to affect recruitment (Bond, 1984; Bond, Vlok & Viviers, 1984; Heelemann et al., 2008), namely fire return interval (pre-fire vegetation age at the time of fire) (n = 21), post-fire age of the vegetation (at the time when recruitment was assessed) (n = 46), parent density (n = 33), post-fire rainfall (over six months post-fire) (n = 46) and species (n = 46).

We used a linear plus rule-based ensemble procedure ‘RuleFit’ with ten-fold cross-validation (Friedman & Popescu, 2008) to determine the importance of the predictor variables listed above in affecting seedling-parent ratio. Predictor importance is expressed relative to the most important predictor and reported on a percentage scale. Predictor effects are shown using partial dependence plots; these show the marginal effect of a predictor on the response variable after taking account of the average effect of the other variables in the model (Friedman & Popescu, 2008). An advantage of using RuleFit compared to a standard regression method is that the most important predictor variables are automatically selected by the method, as are the important interactions (a consequence of the base-learners in the boosting step being decision trees). These are important considerations when (as here) the intention is to determine from the experimental data what the main predictors of the seedling-to-parent ratio are or might be. We subsequently used a generalized linear model (GLM) fitted by quasi-likelihood and the so-called square link (McCullagh & Nelder, 1989; Hilbe, 2011), to test for the significance of the effect of fire season on seedling-parent ratio. Statistical analyses were done using R (R Development Core Team, 2016).

Seed planting experiment

We conducted a seed planting experiment to assess the influence of planting season, intended to simulate fire season (see below; hereafter referred to as an effect of fire season) on germination and survival to one year post-planting of three common overstorey proteoid species (Le, Pm and Pn) in the study area. We conducted the experiment at three spatially well-separated sites: ‘West’ (33.98094°S, 23.20743°E, elevation 312 m), ‘Central’ (33.90880°S, 23.43462°E, 553 m), and ‘East’ (33.96483°S, 24.26432°E, 488 m) (Fig. 1). All three sites occurred at post-fire vegetation ages of >10 years, on gentle north-facing slopes. The experiment thus entailed three factors in a completely crossed design: (i) planting season–summer, autumn, spring and winter; (ii) site–West, Central and East; and (iii) species–Le, Pm and Pn, the most common overstorey proteoids in the area.

At each site, the vegetation in an area of 15 m × 15 m was slashed at ground level and removed to simulate the effect of fire. The germination requirements of proteoids are well-understood and are strongly dependent on moist, cold (1–11°C) conditions (in part provided by removal of the insulating effect of vegetation) rather than on the direct effects of fire (e.g., heat scarification or smoke/ash leachates) (Van Staden & Brown, 1977; Le Maitre, 1990; Midgley & Viviers, 1990; Mustart & Cowling, 1991; Mustart & Cowling, 1993; Brown & Botha, 2004). Areas cleared of vegetation may thus be used to simulate post-fire environments in order to study proteoid recruitment dynamics (cf. Bond, 1984; Midgley, Hoekstra & Bartholomew, 1989). Simulating fire by clearing vegetation furthermore precludes the introduction of unwanted variation in aspects of fires (e.g., fire intensity, which cannot be fully controlled during experimental burning) that are not the focus of the study. We treated the cleared area with a domestic disinfectant (“Jeyes Fluid” with active ingredients being tar acid, washed neutral oil and methanol, diluted at 60 ml per 5 litre of water, as used in commercial farming of proteoids, K Bezuidenhout, pers. comm., 2010; with 30 litres of the solution applied to each site) to simulate the sterilising effect of fire on pathogens, notably the fungi Colletotrichum gloeosporoides and Phytophthora cinnamomi (Botha & Le Maitre, 1992). The cleared area included a 3 m buffer around the perimeter (to reduce edge- and shading effects) with the experimental site in the centre. Flower heads (seed cones) of the study species were harvested from local populations in the vicinity of the study sites one month prior to each of the four planting occasions. Cones were harvested from the current or previous season’s crops and oven-dried at 40°C until seeds were released (Mustart & Cowling, 1993). Apparently viable (plump and unscarred) seeds were hand-sorted (Mustart & Cowling, 1991); about 90 % of such sorted seeds will germinate in controlled conditions (Le Maitre, 1990).

Each of the three experimental sites was divided into twelve plots (of 2.0 m × 2.5 m each) and season allocated randomly to the plots (three plots per season). The seeds were planted on four occasions: in July 2010 (winter), October 2010 (spring), January 2011 (summer) and April 2011 (autumn). One month prior to the planting of seed, regrowth of vegetation was cleared again and disinfectant reapplied to the respective season’s plots. Seeds were lightly pushed into the ground (such that the top parts of seeds were flush with the soil surface or slightly covered with soil; Mustart & Cowling, 1991; Mustart & Cowling, 1993), simulating the habit of Protea seeds to anchor and orientate optimally in the soil by means of specialised hairs (Rebelo, 2001) and the depth of burial by scatter-hoarding rodents (Midgley et al., 2002). Seeds and seedlings were protected from small mammal and bird predation (Bond, 1984; Le Maitre, 1988) by exclosures made from bird mesh (13 mm gauge size, with negligible effects on micro-climate relevant to germination), closed at the top and pegged to the ground. Without exclosures, post-fire seed predation is very high (80% over 15 weeks; Bond, 1984), which would have precluded an experimental assessment of the effects of fire season on seedling recruitment. Seeds were planted in rows such that seeds within and between rows were 50 mm apart (Midgley, Hoekstra & Bartholomew, 1989; Mustart & Cowling, 1993) with 102 seeds of each species planted per plot. A total of 306 seeds of each species were planted per site per season, 1,224 seeds per species per site, and a total number of 11,016 seeds across sites, seasons and species. Germination and seedling survival were monitored during the first week of each month for one year after planting. A standard rain gauge was mounted 1.2 m above the ground at each site and rainfall measured monthly.

Probability of germination and survival to one year (hereafter referred to as ‘recruitment’) was the measure of interest and the focus of data analysis. We explored the effects of season, site and species on recruitment (at one year post-planting; expressed as a proportion of the number of seeds planted per species per plot), using a series of fixed-effects and mixed-effects logistic regression models (detailed in Table S2) using the binomial family and the logistic link, giving regression coefficients that represent log-odds. The random-effects structure of the mixed models best matched the design of the experiment, with plot exclosure nested in season, which was nested in plot, which was nested in site (i.e., 1—Site/Plot/Season/PlotExcl). We assessed the significance of effects using Wald tests (Harrell, 2001; Agresti, 2002). Replicates of planting season only covered one year of study for each level. It might therefore be argued that our results are only relevant to this period. The climatic conditions (rainfall and temperature) that prevailed during the course of the study were, however, within the norm for the area (Fig. S1). We thus argue that our results are generally applicable to the area of study.

Results

Post-fire recruitment surveys

Seedling-parent ratios varied widely (0–43, coefficient of variation 115%) within and among fire seasons, species and regions (Outeniqua vs. Tsitsikamma; Fig. 1) (Fig. 2). The RuleFit model fitted the data well (variance explained, 97.7%; normalised root-mean-square error, 0.15; normalised standard deviation, 0.99). The model showed the most important variables affecting recruitment success to be species (estimated relative importance averaged over all predictions, 100%), parent density (56%), post-fire vegetation age (55%) and fire season (39%), while post-fire rainfall (22%) and fire return interval (11%) were unimportant (Fig. 3). Autumn and spring fires resulted in better recruitment (of species combined) than winter and summer fires. Recruitment was negatively related to parent density at densities of <6,000 parents/ha and positively related to the post-fire age (>26 months) of the vegetation at the time of assessment. According to the GLM, fire season was not significant at the 5% level in affecting post-fire recruitment success (F3,42 = 2.53, P = 0.07; detailed model output in Table S3).

Figure 2 Recruitment success after fires in different seasons.

Recruitment success, expressed as seedling-parent ratio, is shown for different proteoid species (Le, Leucadendron eucalyptifolium; Lu, L. uliginosum; Pe, Protea eximia; Pm, P. mundii; Pn, P. neriifolia) after fires in different seasons. We distinguish between two regions, (A) Outeniqua and (B) Tsitsikamma, in the study area.

Figure 3 Partial dependence of post-fire recruitment success on predictor variables.

Plots of partial dependence of post-fire recruitment success (measured as seedling-parent ratio) on predictor variables (A–F, in decreasing order of importance) as modelled by a linear plus rule-based ensemble procedure (RuleFit): (A) species (Le, Leucadendron eucalyptifolium; Pe, Protea eximia; Pm, P. mundii; Pn, P. neriifolia), (B) parent density, (C) post-fire vegetation age at the time of assessment, (D) fire season (Wi, winter; Sp, spring; Su, summer; Au, autumn), (E) post-fire rainfall (during six months post-fire) and (F) fire return interval.

Seed planting experiment

Total rainfall during the 21-month course of the experiment was comparable among sites (West 1,315 mm, Central 1,371 mm, East 1,397 mm) and not indicative of a strong rainfall gradient. Overall, 38% of planted seeds (Le 20%, Pm 45%, Pn 50%) germinated, and 84% of these germinants (Le 70%, Pm 86%, Pn 90%) survived. Germination was limited to the cooler months (March/April–November) with seeds planted in spring, autumn and winter mostly germinating within two months post-planting (Fig. S2). In contrast, seeds planted in summer only germinated four to five months post-planting with the advent of cooler conditions. Additionally, a small proportion of winter- and spring-plantings germinated during their second cold season post-planting. We observed no obvious association between monthly rainfall and the timing of germination (or mortality) in our experiment (Fig. S2).

Recruitment differed significantly among planting seasons, sites and species, with Le recruitment being poorer than that of Pm and Pn (Fig. 4). Recruitment pooled across sites and species was highest in winter-plantings (37.9 ± 4.0%, mean ± SE), decreasing through autumn-(35.5 ± 3.9%) and summer-(29.8 ± 2.7%) to spring-plantings (26.3 ± 3.4%) (Fig. S2). Significant interactions occurred among the experimental factors (Table 1; Fig. 4), i.e., the effect of planting seasonality on recruitment success was not consistent among species within sites, nor among sites within species. Recruitment responses at the western and central sites were more similar than at the eastern site.

Table 1 Type II Wald χ2 tests of the effects of planting season, species, and site on recruitment (measured as survival at one year post-planting as a proportion of seeds planted) based on a generalized linear mixed-effects logistic regression model.

	Wald χ2	df	Pr (>χ2)	
Season	42.7	3	<0.001	
Species	713.2	2	<0.001	
Site	45.7	2	<0.001	
Season × Species	105.4	6	<0.001	
Season × Site	34.5	6	<0.001	
Species × Site	67.2	4	<0.001	
Season × Species × Site	177.7	12	<0.001	

Figure 4 Predicted effects of planting season, species (Le, Leucadendron eucalyptifolium; Pm, Protea mundii; Pn, P. neriifolia), and site (West, Central, East) on the probability of recruitment (survival at one year post-planting) based on a generalized linear mixed model (see Table S2 for model outputs).

Bands show asymptotic 95% confidence intervals.

Discussion

Germination cues: moisture and temperature

Levels of proteoid germination (20–50%) in our seed planting experiment were comparable to those in other field studies (c. 10–60%, Cowling & Lamont, 1987; 24%, Midgley, Hoekstra & Bartholomew, 1989; 45–80%, Mustart & Cowling, 1993) and under optimal laboratory conditions (30–60%, Van Staden & Brown, 1977). Distinguishing between fertile and infertile seeds is problematic in Leucadendron, unlike in Protea (Van Staden & Brown, 1977). Ineffective sorting of Leucadendron eucalyptifolium seeds may thus have accounted for their comparatively poor germination in our experiment, although poor recruitment does not appear to be the norm in Leucadendron in the field, as seen from our post-fire recruitment surveys.

We asked whether the establishment of obligate reseeding shrubs would be constrained by rainfall in aseasonal environments as elsewhere in Mediterranean-climate shrublands where seasonal droughts are a feature (Bond, 1984; Lamont, Connell & Bergl, 1991a; Moreno et al., 2011; Céspedes et al., 2012; Mustart et al., 2012). Proteoids show a summer drought-avoiding dormancy in many areas (Deall & Brown, 1981; Bond, 1984; Midgley, Hoekstra & Bartholomew, 1989) with germination following a temperature plus moisture cue (Van Staden & Brown, 1977) which is met by the cold and wet conditions of winter under Mediterranean climates (Cowling & Lamont, 1987; Le Maitre, 1988; Le Maitre, 1990; Mustart & Cowling, 1991). However, we observed no obvious association between (i) monthly rainfall and the timing of germination (or mortality) in our experiment, or (ii) post-fire rainfall and recruitment success in our post-fire surveys, suggesting that post-fire rainfall per se is seldom limiting to recruitment success in the study area. We could, however, not assess potential effects of fire season and post-fire climatic conditions on very early mortality of seeds exposed on the soil surface, due to constraints of monitoring frequency and the need to secure seed positions through ‘planting’ in our experiment.

In the semi-arid Swartberg Mountains (inland of the study area, where rainfall is also largely aseasonal, but where summer droughts are more severe due to higher evapo-transpiration associated with higher temperatures and lower humidity; Seydack, Bekker & Marshall, 2007) germination of proteoids was strongly correlated with temperature but not with monthly rainfall (Midgley, Hoekstra & Bartholomew, 1989). We observed a delay in germination following summer-planting which appeared to be due to the absence of low temperatures (minimum monthly temperature <10°C) that are typically needed to stimulate germination in proteoids (5°C, Van Staden & Brown, 1977; 10°C, Mustart & Cowling, 1991) and Erica (Moreno et al., 2011) elsewhere. In the aseasonal shrublands of southeastern-Australia, ambient temperature also strongly controls germination, with high summer temperatures presumably imposing secondary dormancy on seeds irrespective of rainfall (Bradstock & Bedward, 1992). During mid-summer, germination thus appears to be prevented or to fail (germinants succumbing to desiccation; Mustart et al., 2012) due to moisture deficits associated with high temperatures rather than an absence of rainfall in aseasonal climates, as opposed to the combination in Mediterranean climates (Deall & Brown, 1981).

Effect of fire season on recruitment

We found that fire season or its simulation had little consistent effect on post-fire recruitment success, which is in strong contrast to the consistent responses of proteoids to fire season in many southern hemisphere shrublands (Midgley, 1989). Under more seasonal (winter rainfall and summer drought) and less equable climates, spring and summer fires resulted in increased pre-germination mortality of proteoid seed due to extended post-fire exposure to predation and decay (Bond, 1984; Bond, Vlok & Viviers, 1984; Enright & Lamont, 1989; Midgley, Hoekstra & Bartholomew, 1989). Under a regime of evenly distributed rainfall, however, favourable conditions for germination after spring or summer fires are likely to occur sooner or more regularly than under winter-rainfall or semi-arid regimes. Under aseasonal climates, the germination delay is thus contracted and season of fire presumably less influential on post-fire recruitment success. It is unlikely that predator exclosures used in this study would have materially influenced our findings, as the effect of seed predation on disparate recruitment success among fire seasons observed in winter rainfall fynbos relate more to the duration of seed exposure to predation (Bond, 1984; Le Maitre, 1988) than to the seasonality of seed exposure and seasonality of rodent consumption (Holmes, 1990; Rusch, Midgley & Anderson, 2014). Seed consumption by rodents is furthermore expected to be less seasonal in eastern fynbos where plant phenology (and thus seed production) is less seasonal (Cowling, 1987).

Recruitment success was not consistently superior after fire in any particular season in our study. Trends from our post-fire recruitment surveys (peaks in recruitment success after autumn and spring fires) differed from those of the seed planting experiment (peaks after winter and autumn fire simulation), while results furthermore varied greatly within and among species and sites (∼habitat types with diverse soils, slopes and aspects) in both these datasets. Collectively, this suggests that seasonal patterns in post-fire recruitment are weak in eastern coastal fynbos. In eastern inland fynbos, in sites drier than our coastal ones, Heelemann et al. (2008) observed peaks in recruitment after autumn and spring fires, similar to results from our post-fire recruitment surveys, and explained these on the basis of the bimodal (spring-autumn) rainfall regime of the area. However, we question whether this simple relationship adequately explains, and provides evidence of, seasonality in post-fire recruitment in eastern coastal fynbos, particularly in light of a lack of correlation between germination and rainfall in both eastern coastal fynbos (see above) and further inland (Midgley, Hoekstra & Bartholomew, 1989). Moreover, rainfall in the study area is not strictly bimodal and may be more appropriately described as aseasonal with marked variation in seasonality among years (cf. Fig. S1).

We maintain that seasonal patterns in post-fire recruitment are weak in eastern coastal fynbos; good (or poor) recruitment may be expected at any time of the year and may vary considerably between years and habitat types. A weak seasonal response in recruitment is plausible under an equable, coastal climate with year-round rainfall and is in accordance with the lack of seasonality recorded both in weather conditions conducive to fire (Kraaij, Cowling & Van Wilgen, 2013a) and historical fire occurrence (Kraaij et al., 2013b) in the area. In the analogous aseasonal shrublands of southeastern-Australia, the effects of fire season on recruitment are equally unpredictable, given the high level of year to year variation in seasonal rainfall (Bradstock & Bedward, 1992). These authors suggested that, in the longer term, the timing of fire relative to sequences of wet and dry years may be just as important as fire season in its effect on proteoid populations. Accordingly, the interaction between rainfall variability and fire season was shown to disparately affect recruitment of different species of reseeding shrubs in the Mediterranean Basin (Moreno et al., 2011).

Effects of other factors on recruitment

Our results suggest that recruitment may also vary according to species, the density of parent plants, and the post-fire age of the vegetation at the time of assessment. Large variation in post-fire recruitment, as observed in our study (Fig. 2), is characteristic of fynbos proteoids (Van Wilgen & Viviers, 1985; Cowling & Gxaba, 1990), even within favourable fire seasons (post-summer/autumn fire seedling-parent ratios of 12–19, Bond, 1980; 0–21, Bond, Vlok & Viviers, 1984; 3–15, Van Wilgen & Viviers, 1985; 0–9, Le Maitre, 1988), and may be caused by a variety of factors. In our study, individual species differed in their recruitment responses (Bond, Vlok & Viviers, 1984; Midgley, Hoekstra & Bartholomew, 1989), which may be related to wide variation (among species and/or habitats) in post-emergence desiccation tolerance (Mustart et al., 2012). Large variability in regeneration within and between species and fire events (cf. Moreno et al., 2011), suggests that generalisations based on studies of single species or fires should be treated with considerable caution.

In our study, parent density had a greater effect on recruitment success than fire season had. Other studies have found proteoid parent density either to have no effect (Cowling & Gxaba, 1990), or more (Le Maitre, 1988) or less (Bond, Vlok & Viviers, 1984; Midgley, 1989; Esler & Cowling, 1990) effect, than that of fire season. Negative effects of parent density on recruitment have been ascribed to suppressed seed production (Bond, Vlok & Viviers, 1984), although evidence is conflicting at the individual and population levels (Esler & Cowling, 1990; Treurnicht et al., 2016).

Recruitment success in our study increased with post-fire vegetation ages (at the time of assessment) exceeding 26 months. This does not support the notion of increases in seedling mortality with post-fire vegetation age (Bond, Vlok & Viviers, 1984). Instead, decay of parent skeletons may have resulted in undercounting of parents (and thus overestimation of seedling-parent ratios) in older post-fire ages. Alternatively, small size of seedlings in very young post-fire ages may have resulted in their being undercounted. Restricting the observation window to 1–2 years post-fire should reduce this source of noise in the data. Fire return interval may also affect recruitment success (Bond, 1980; Treurnicht et al., 2016) through its effect on seed availability (related to plant maturation rates; Muir, Vesk & Hepworth, 2014), but had no effect in our study as we deliberately excluded data of short (<7 year) interval fires, known to inhibit recruitment (Kraaij et al., 2013c).

Management implications

Prescribed burning is seen as an important management option in fire-prone shrublands globally (Van Wilgen, Richardson & Seydack, 1994; Morrison, Buckney & Bewick, 1996), but its use is constrained by many factors, including the need to burn within acceptable limits of season, frequency and intensity (Bradstock, Williams & Gill, 2002; Van Wilgen et al., 2011). The weak and varied response of proteoid recruitment to fire season implies that burning does not have to be limited to particular seasons in eastern coastal fynbos, and this would remove at least one constraining factor, which should improve the chances of carrying out successful burns. However, various other constraints on fire management remain. Fire return intervals should allow for adequate seed production in slow-maturing obligate reseeders to ensure post-fire regeneration (Kraaij et al., 2013c; Muir, Vesk & Hepworth, 2014). Fire intensity needs to be sufficiently high to stimulate seed release and germination in serotinous (Bradstock & O’Connell, 1988; Midgley & Viviers, 1990) and large or hard-coated, soil-stored seeds (Jefferey, Holmes & Rebelo, 1988; Bond, Le Roux & Erntzen, 1990; Knox & Clarke, 2006), but not too extreme that all seeds of fine-seeded species in the surface layers of the soil be killed (Bond, Honig & Maze, 1999). Additionally, there is evidence that variation in fire regimes is necessary to maintain plant diversity in the landscape (Thuiller et al., 2007; Gill & McCarthy, 1998), and particularly in an unpredictable, aseasonal environment.

Ecological requirements of fire regimes furthermore have to be traded off with the need for safety of human lives and assets (commercial timber plantations, in particular in the study area; Kraaij, Cowling & Van Wilgen, 2011), which often present considerable management challenges (Morrison, Buckney & Bewick, 1996; Van Wilgen, Forsyth & Prins, 2012). The incidence of weather conditions that would meet both the ecological need for fire intensity and human needs for fire safety is typically low in fynbos environments (c. 10% of days annually; Van Wilgen & Richardson, 1985). Implementation of further restrictions based on fire season (arising from research suggesting that fynbos recruitment is highly seasonal; Bond, Vlok & Viviers, 1984; Van Wilgen & Viviers, 1985) made prescribed burning of fynbos at a large scale unattainable (Van Wilgen, 2009). The lack of a seasonal restraint on burning in eastern coastal fynbos therefore has significant and encouraging management implications in affording more flexibility for fire management in this area, although the ecological need for sufficient fire intensity remains.

Even though wildfires almost completely dominated the recent fire history of the area (Kraaij et al., 2013b), prescribed burning remains necessary: (i) in key locations to reduce the risk of fire spreading from fynbos to adjacent timber plantations (Kraaij, Cowling & Van Wilgen, 2011); (ii) along the wildland-urban interface (Radeloff et al., 2005; Van Wilgen, Forsyth & Prins, 2012); (iii) as a tool in the management of invasive alien plants (Roura-Pascual et al., 2009; Van Wilgen et al., 2016); and (iv) to rejuvenate fragments of fire-dependent vegetation where ignition sources have been reduced or eliminated by transformation of the surrounding landscape. Our findings suggest that prescribed burning may be done in these instances during any season within a framework of adaptive management (Van Wilgen et al., 2011). Managers furthermore do not have to allocate large amounts of resources to fight wildfires that are burning in the ‘wrong’ season and may conduct back-burns to contain wildfires during any season. In conclusion, because the seasonal occurrence of fires may vary over the geographical range of a particular vegetation type, the responses of the vegetation to fires in different seasons clearly need to be documented across the geographical extent of the vegetation type to refine guidelines for fire management.

Supplemental Information

Figure S1 Weather conditions during the study period compared to long-term record

Mean monthly minimum and maximum temperatures and rainfall compared between the study period of the seed planting experiment and the remainder of the long-term record available for the nearest and most central weather station (Plettenberg Bay, 1993–2013). Bars show minimum and maximum recorded figures for long-term data.

Click here for additional data file.

Figure S2 Progression in time of germination and mortality of planted proteoid seeds

Live recruits of three Proteaceae species (Le, Leucadendron eucalyptifolium; Pm, P. mundii; Pn, Protea neriifolia) observed within the first week of each month, expressed as a percentage of seeds planted under predator-exclosures after clearing of above-ground vegetation (simulating fire) in four seasons at three sites (West, Central, East). Zero values at the start of each series mark planting occasions for each of the austral seasons.

Click here for additional data file.

Table S1 Locations of post-firerecruitment survey sites

The number of belt transects (each 2 m × 30 m), the season of fire, and the proteoid species (Le, Leucadendron eucalyptifolium; Lu, L. uliginosum; Pe, Protea eximia; Pm, P. mundii; Pn, P. neriifolia) surveyed at each site are shown .

Click here for additional data file.

Table S2 Output of logistic regression models

∗ 0 outside the confidence interval

ǂ Numbers in parentheses after the estimated regression coefficients (βs) are asymptotic 95% confidence intervals. Models fitted using treatment contrasts. The (Intercept) therefore represents the log-odds of recruitment for Species = Le during Season = Winter at Site = West. For the mixed model, Var in the lower part of the table gives the variance of intercepts in groups of records identified by the referenced group in the random-effects structure. That is, the entry Plot:Site (Intercept) refers to the variance in the 12 intercepts of Plot within each level of Site in the second-level grouping of the random-effects structure (1—Site/Plot/Season/PlotExcl).

Click here for additional data file.

Table S3 Generalized linear model output

Click here for additional data file.

Data S1 Raw data of seed planting experiment and post-fire recruitment field surveys

Click here for additional data file.

We thank Johan Huisamen, Johan Baard and George Sass for field assistance; Jan Vlok, Yvette Van Wijk, Karien Bezuidenhout, Penny Mustart, Andrew Latimer, Anthony Hitchcock and Tony Rebelo for advice with the germination experiment; Andrew West for sharing seedling-parent ratio survey data for two sites; Jeanette Pauw for initial statistical advice; Thertius Notley for weather records of Keurboomsrivier Plantation; the South African Weather Service for weather records; and South African National Parks, Eastern Cape Parks and Tourism Board, and Mountain to Ocean Forestry for permission to work on land under their management. David le Maitre and two anonymous reviewers provided useful comments which led to the improvement of this manuscript.

Additional Information and Declarations

Competing Interests

Author Contributions

Field Study Permissions

Data Availability

Richard M. Cowling is an Academic Editor for PeerJ. The authors declare there are no competing interests.

Tineke Kraaij conceived and designed the experiments, performed the experiments, analyzed the data, contributed reagents/materials/analysis tools, wrote the paper, prepared figures and/or tables, reviewed drafts of the paper.

Richard M. Cowling and Brian W. van Wilgen conceived and designed the experiments, wrote the paper, reviewed drafts of the paper.

Diba R. Rikhotso performed the experiments, contributed reagents/materials/analysis tools, reviewed drafts of the paper.

Mark Difford analyzed the data, contributed reagents/materials/analysis tools, wrote the paper, prepared figures and/or tables, reviewed drafts of the paper.

The following information was supplied relating to field study approvals (i.e., approving body and any reference numbers):

South African National Parks gave approval for a research project conducted within the Garden Route National Park (the main author was a permanent employee at the time).

Eastern Cape Parks and Tourism Board gave permission for conducting field experiment and surveys on land under their management.

The following information was supplied regarding data availability:

The raw data has been supplied as Data S1.

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
