# Peer review of "Vegetation responses to season of fire in an aseasonal, fire-prone fynbos shrubland"

_PeerJ, doi:10.7717/peerj.3591_

## Round 0.1 · original submission · Major Revisions

All three reviewers agreed that this paper is interesting and well-suited for publication in this journal. I agree with that assessment and think it is an exciting addition to the fire ecology literature. All three reviewers suggested the paper could be considered for acceptance following minor corrections. It was, however, my assessment that the total extent of changes requested by the reviewers meant that major revisions are needed. This was primarily because the reviewers determined there was a need for considerable additional detail in the methods and presentation of the results as well as some additional analyses.

In revising your paper I would draw your attention to the following key issues to consider:
1) Additional detail is needed in the Methodology especially with regards to the seed-planting experiment and data analysis methods. Please ensure your approaches are described in adequate detail, well-justified and that any potential limitations are made clear
2) Two reviewers requested further explanation of how you determined what constituted a "sufficient" sample size for your counts
3) One reviewer expresses concerns about the methods used to assess rainfall across your sites. This does appear to be a critical issue and should be explicitly addressed
4) Please ensure statistical results are reported in full if not in the main paper then in the supplementary material
5) Ensure you address the reviewer's requests for supplementary analyses or additional information on the analyses already completed

Finally, one reviewer recommended providing some additional global context for your study with regards to fire effects on shrublands. I agree that would be helpful and that it could be briefly addressed in the introduction and discussion. For conflict of interest reasons I have, however, removed the references suggested by Reviewer 2 from their notes.

Reviewer 1 ·

Basic reporting

Some additional reporting of results would benefit the manuscript. Specifically, for the seed planting experiment, please report model outputs in addition to Fig. 3. For the recruitment survey, please also report coefficients for the GLM. With regard to the variables included in the analysis of the recruitment survey, while it is interesting to see the relative variable importance reported by the partial dependence plots in Fig. 2, perhaps the actual relationships could be reported in the supplementary materials (bar charts, scatterplots, etc, depending on the predictor variable).

I would also add a summary table of each species actual seedling counts (from both studies), perhaps averaged by season of fire to supplementary materials.

Authors should report on recruitment patterns in unburned areas in this region, if possible (even if anecdotal), to provide a reference point for fire effects.

General comment related to Introduction: are management protocols developed in the west actually being applied to the eastern coastal region (as implied lines 66-67)? If so, you might state what these practices currently are, giving more context to your recommendations in the Discussion.

Also, in the Abstract, state how many focal species there were.

Experimental design

The research questions were well defined, relevant, and meaningful. However, I would like the authors to include additional information on their methods, which I think will help improve the interpretibility of the results greatly.

line 119: “proportion of summer rain” – perhaps reword to “proportion falling as summer rain” for clarity
line 135: was each site surveyed only once?
line 143: provide the range of number of transects surveyed per site in addition to the mean and SE reported
line 146: was each species sampled at each of the four burn times?
line 143-144: was only one species surveyed per transect?
line 144: explain what “sufficient numbers” means
Several species were surveyed in a given site- I’d consider adding a random site-level effect, and if applicable (I can’t determine by current level of description of survey methods) a transect within site effect if multiple species were surveyed in the same transect.
line 157: post-fire rainfall variable- be more specific what this means (total?)
What were the criteria for site and transect selection?
line 224: state the random effects structure
lines 225-229: consider moving this to the Discussion
Figure 1 – overlapping points are very hard to see. Consider jittering the points or increasing the contrast by using different colors or shades for each species.
Figure 1 – Species ‘Lu’ seems to be missing
line 387-398: sentence wording is unclear as written

Validity of the findings

No further comments beyond suggestions above.

Reviewer 2 ·

Basic reporting

See comments to the authors

Experimental design

See comments to the authors

Validity of the findings

See comments to the authors

Comments for the author

The objective of this paper was to determine the effect of fire season on recruitment of a number of proteids of eastern coastal fynbos of South Africa characterized by an aseasonal climate. To do that, the authors sampled recruitment in twenty sites that burned at different times of the year, among other. Additionally, they planted seeds at various times of the year and monitored their recruitment at a number of sites. The paper is clear, concise and well written, the scientific questions are clearly stated, the analysis is robust and, for the most part sound (but see later), the discussion is well supported by the data and the conclusions and implications for management are most relevant. The paper deserves being published. I had a number of observations that need to be considered.
It was unclear to me how the siteXspecies were obtained. Since no table was provided with the number of transects that were made at each site, it appears as if transect within site were considered as samples, not subsamples. This may affect the overall results, owing to changes in the degrees of freedom, and is something that needs to be clarified.
In the introduction, at the setting of the problem it is stated that (lines 61/62) ” Empirical evidence indicates that season of fire can affect species abundance and floristic composition in fire-prone Mediterranean-climate shrublands (Bond 1984; Enright & Lamont 1989; Midgley 1989).” Based on the references provided it seems that the authors are referring to southern hemisphere mediterranean-climate shrublands. The literature review did not include any reference for northern-hemisphere shrublands. Definitely, this sentence does not fully apply to the latter (see some references below). Please, reword for clarity and, if you deem appropriate to expand your text beyond the southern hemisphere, consider citing the relevant paper that have been recently published (I include a few of them for your information). I would encourage the authors to do that in order to capture a wider attention towards this issue.
A minor point: a counted the sites in Fig S1 and found only 19, but maybe there are overlaps that I could not. I call the attention to the authors for in case there is an omission that they want to correct.

·

Basic reporting

Line 76 – decline should be declines
Line 77 – replace owing to with because
Line 78 – replace being and being with is and are respectively
Line 177 – I have heard of balanced designs but not crossed designs – please explain
Line 182 – is should be are
Line 187 – 2nd sentence – what It refers to is not clear
Lines 276-278 – this is phrased as a statement not a question
Line 337-8 – This is a very awkward sentence. Simply say that year to year variability needs to be examined still.
Line 396 – this is an awkward sentence. I suggest “Even though wildfires almost completely dominated….”

Experimental design

Lines 158-167 I am not familiar with this specific multivariate technique but recognise that statistics, like other sciences, is continually improving its techniques. I assume that it is appropriate for this kind of analysis but I would like a better explanation/rationale for its use as opposed to other forms of multivariate regression/analysis.
Lines 189-193 – Two things. I would like to know if the effects of this treatment have been tested to see whether they have a negative effect on germination and seedling establishment or if there is reason to assume that it does not. I would like to know if this treatment (e.g. the given concentration) has been shown to be effective in controlling the pathogens they were concerned about.
I am not sure why the seedling to parent counts were done using a different sampling design (belt transects) from that used in most of the previous studies (circular plots). It would be useful to have an explanation of the rationale and why the particular dimensions were chosen. You state that the number of transects varied between the sites so that they could record “sufficient” numbers of parents and seedlings. You need to state clearly what you regarded as “sufficient” and why.

Validity of the findings

Main comments
1) The study examines Proteaceae regeneration in two ways – field observations and seedling establishment, the latter allowing them to assess seed emergence as well. I have concern about the seedling establishment experiment. This involved seed planting which is not the normal situation for seeds of serotinous species which typically germinate on the surface. This is important because seed wetting and drying is more rapid on the surface than in the soil. This, in turn, makes the potential for seed mortality before the radicle enters the soil and starts absorbing moisture much greater. It can also affect the availability of water to hydrate the seed for long enough for germination to begin. Which means that even short-dry periods between rainfall events can have significant impacts on germination and survival during emergence and establishment (the ability to tolerate desiccation has been noted by Mustart - cited in the discussion in this MS but should come into the introduction). The vulnerability is offset by to some degree by the fact that the seeds are wind dispersed and tend to settle in hollows or against obstructions which can provide some protection or trap moisture. I am not arguing that the seed planting invalidates the results, only that the vulnerability to drying out is a factor you need to discuss explicitly in the MS.
2) This comment relates to the one above. The study examined rainfall in the first 6-months after the fire as a factor potentially explaining recruitment success. Given the above, and the fact that the establishing seedlings are far more likely to die of desiccation than of excessive moisture in these well-drained soils, I think that it could be more insightful to examine the rainfall records for the lengths of the periods between rainfall events (i.e. successive days with no rain vs some rain). Even though the rainfall is more-or-less evenly distributed, the higher temperatures in summer would tend to dry out the soil surface more rapidly between rainfall events and the rainfall events may be of shorter duration. These factors would influence seed hydration and thus seed germination, emergence and establishment and may explain the delayed germination in plantings in summer (the explanation in lines 297-8 is not correct – high temperatures result in more rapid drying out of soils and seeds which inhibits germination; alternatively seed germination may be inhibited directly by high temperatures with the dryness being incidental?). The wetting and drying also depend on the duration and intensity of the solar radiation which would differ on northern and southern aspects, or east and wet aspects, both for the mountain ranges as a whole and slopes at the sampled sites. You should at least note these factors as potential explanations of the observed variability and evaluate the points they make in their discussion in this light.
3) You should make it clear that all these sites were on the southern (wetter) side of the mountain ranges. Patterns may be different on the northern (dryer) side of the mountain ranges. This is important because the fires are often associated with Berg winds which “force” fires from the northern side to burn across to the southern side so that fires occurrences on the northern and southern side are not independent and the plants may have adapted to this. Decisions about when to do prescribed fires should take these observed patterns in fire occurrence into account.
4) Using only one rain gauge record to represent sites spread over an east-west distance of roughly 200 km in an area with very variable rainfall needs some justification especially given that this region is an area of overlap between the summer and winter rainfall circulation systems rather than a system of its own. Thus the observed rainfall in any season and year depends on the relative strengths of these interactions, which explains why it is so variable. I doubt that one record, even though it is centrally located, would be enough to accurately represent, for example, the critical dry periods I have noted above.
5) In the abstract and in the text (near the end) you state that proteoid recruitment in this study was found to be “inconsistent” and “random”. In my experience biological processes like germination and growth are deterministic in the sense that once they are initiated they proceed at a certain pace and require a certain timespan. The same with establishment, where it takes time for seedling emergence and rooting. These plants have interacted with the selection pressures imposed by temporal patterns of temperature and rainfall; these are consistent (or predictable) enough to allow selection and evolution to take place so that species evolve, for example, germination cues that maximise recruitment. My point is that I do not think the terms “inconsistent” and “random” are appropriate. There may be variability that cannot be explained using the factors that were considered in this study, but that does not make the underlying dynamics “random” or “inconsistent”.
6) In the study of seedling parent ratios it may be worth examining whether the relationships between post-fire age and recruitment differs between sites which had not yet gone through a prolonged dry period (summer?) from those that had.
7) Lines 378-381 – Fire intensity is a double–edged sword. As noted it needs to be high enough to result in a “clean” burn to create conditions conducive to the germination of large soil-stored seeds. However, study by Bond (et al. 1999, Oecologia 120, 132) argued that intense fires can kill all the seeds in the surface layer, something that can be detrimental for fine-seeds species such as Ericaceae, a point that needs to be noted.
Other comments
I am not sure why figures S1-3 were not included in the MS, especially S1. I found that they were insightful and having them integrated into the MS would make it much easier for the reader to refer to them when they are mentioned in the text.
Line 101-2 I think van Wilgen and Viviers may have hypothesised that recruitment success is unrelated to season but my recollection is that they rejected that hypothesis. I suggest that this is not a sound reason to adopt the same hypothesis.

Comments for the author

This is a generally well presented and written, logically organised paper and I found it easy to read. It is also clear that it involved a lot of work in gathering and sorting seeds, and tracking the outcomes as well as in the post-fire surveys. I do think there are some points of interpretation you need to consider as possible explanations of the patterns you have observed. There are also some minor comments and edits under the respective sections.

---

## Round 0.2 · Minor Revisions

Reviewer #3 has a few comments that you may wish to address. I think your use of RuleFit is ok. Different statistical approaches could also suitable, but just one approach should be used here. In addition, I have the following comments;

L 68 reseeding shrubs muirare often [what does it mean?]

L 258 How did you decide that over-fitting did not occur? Or, if no objective criteria were used, I would simply drop that term.

L 328-366 My understanding is that your experiment (and related interpretations here) was based on caged seeds and seedlings. Such conditions are not natural, so I am left wondering whether your conclusions might have been very different if seeds had not been caged. A short discussion of caging would be beneficial somewhere in this section.

L 328-366 A second point that could merit discussion in this section is that the glm P-value for effect of season was 0.07. The P-value; strongly affected by sample size. I wonder if is a bit risky (type 2 error) to recommend that managers dismiss seasonal effects given the observed p-value? You may be able to refer to the graphs and tables to show that the pattern at your study site was clearly not close to the expectation at more seasonal sites, even if a larger study might have detected P < 0.05 for effect of season.

L 436 This concluding “paragraph” is actually a single sentence, which is not a paragraph. Please split into at least 2 sentences if you want this as a separate paragraph.

Reviewer 1 ·

Basic reporting

The authors have taken care to address all issues I raised during the initial review related to basic reporting. In particular, clarity on methods (both data collection and analysis) has been enhanced.

Experimental design

No comment.

Validity of the findings

No comment.

Comments for the author

The authors have adequately addressed my concerns and those raised by fellow reviewers, especially those related to clarity of methods, analysis, and data presentation. I believe this represents an interesting and timely contribution to the fire ecology literature in MTEs, particularly of the southern hemisphere.

·

Basic reporting

I am satisfied with the rebuttal but do have some further comments (see the other sections) which I believe can be dealt with between yourselves and the editor(s) concerned.

Experimental design

I accept the point the RuleFit may have features that facilitate selection of the best variables as predictors. However there are other, arguably more traditional, methods of variable selection (including interactions) which are also very effective and have been used for decades. I cannot think of any cases where the data do not determine which are the best predictors! So neither explanation is really convincing but this is not a make or break issue. I leave it up to the editor to decide whether they agree or not.
The issue of the Jeyes has been satisfactorily explained but I think you should note in the text that this is a standard method in commercial farming of proteas.

Validity of the findings

The issue of the periods without rain has, apparently, been misunderstood. I was not asking about the total number of dry days over the 6months, but about the lengths of the periods between rain days (i.e. the dry spells or number of consecutive days without rain) during those 6 months. Long dry spells can have big impacts, especially soon after germination. This issue was raised by Bond and Van Wilgen in their 1996 Fire and Plants book and by Cowling et al. 2005 Global Ecology and Biogeography 10.1111/j.1466-822X.2005.00166.x. This point has recently been made by Slingsby et al. PNAS 2017 doi: 10.1073/pnas.1619014114 who noted the effects of increases in dry spells and high temperatures on regeneration.
On the issue of only using rainfall at Plettenberg Bay, the explanation that is is "the most rigorous way” is not in really true; it was a pragmatic choice based on it being deemed to be representative by virtue of its central location. Not so? I am not entirely happy but will not pursue the point further.

Comments for the author

On the issue of south slopes I suggest extending the sentence to read: “size and severity of fires and their spread from the northern to the southern slopes”
On the “random”. I have suggested alternate wording for the Abstract (see below) which could also be carried across to the discussion.
Editorial points – tracked changes MS where I picked up issues
Line 24 reword: “recruitment responses are not related to the season of the fire”
Line 26 reword: “avoiding dormancy is limited and has less influence on variation in recruitment success”
Line 27 reword: “The absence of response to”
Lines 362 reword: “may be just as important as”
Lines 364-366: this sentence does not follow from the previous statement. Do you mean “In contrast?” rather than accordingly

---

## Round 0.3 · accepted · Accept

The authors have made minor revisions suggested by reviewers. I think the manuscript is now acceptable.